# Using interpretable machine learning to predict bloodstream infection and antimicrobial resistance in patients admitted to ICU: Early alert predictors based on EHR data to guide antimicrobial stewardship

**Davide Ferrari**[1,2]*, **Pietro Arina**[3], **Jonathan Edgeworth**[2], **Vasa Curcin**[1],
**Veronica Guidetti**[4], **Federica Mandreoli**[4], **Yanzhong Wang**[1]

1 School of Life Course and Population Sciences, King's College London, London, United Kingdom, 2 Centre for Clinical Infection & Diagnostics Research, St. Thomas' Hospital, London, United Kingdom, 3 Bloomsbury Institute of Intensive Care Medicine, University College London, London, United Kingdom, 4 FIM Department, University of Modena and Reggio Emilia, Italy

* davide.ferrari@kcl.ac.uk

**Data Availability Statement:** In this work we used anonymized, yet sensitive data of living patients

## Abstract

Nosocomial infections and Antimicrobial Resistance (AMR) stand as formidable healthcare challenges on a global scale. To address these issues, various infection control protocols and personalized treatment strategies, guided by laboratory tests, aim to detect blood-stream infections (BSI) and assess the potential for AMR. In this study, we introduce a machine learning (ML) approach based on Multi-Objective Symbolic Regression (MOSR), an evolutionary approach to create ML models in the form of readable mathematical equations in a multi-objective way to overcome the limitation of standard single-objective approaches. This method leverages readily available clinical data collected upon admission to intensive care units, with the goal of predicting the presence of BSI and AMR. We further assess its performance by comparing it to established ML algorithms using both naturally imbalanced real-world data and data that has been balanced through oversampling techniques. Our findings reveal that traditional ML models exhibit subpar performance across all training scenarios. In contrast, MOSR, specifically configured to minimize false negatives by optimizing also for the F1-Score, outperforms other ML algorithms and consistently delivers reliable results, irrespective of the training set balance with F1-Score .22 and .28 higher than any other alternative. This research signifies a promising path forward in enhancing Antimicrobial Stewardship (AMS) strategies. Notably, the MOSR approach can be readily implemented on a large scale, offering a new ML tool to find solutions to these critical healthcare issues affected by limited data availability.

and it is not available for public access. Access to the data is subject to appropriate checks and vetting and researchers can refer to the contact of the database manager Finola Higgins (finola. higgins@kcl.ac.uk).

**Funding:** The author(s) received no specific funding for this work.

**Competing interests:** The authors have declared that no competing interests exist.

## Author summary

This study confronts the global healthcare challenges posed by hospital-acquired infections and antibiotic resistance. It introduces an innovative machine learning approach known as Multi-Objective Symbolic Regression (MOSR), designed to predict bloodstream infections and evaluate antibiotic resistance risks using readily available clinical data from intensive care unit admissions. Unlike conventional models, MOSR consistently outperforms its counterparts, delivering reliable results even when faced with data imbalances. This advancement holds significant promise for enhancing Antimicrobial Stewardship (AMS) strategies, potentially curbing the unnecessary use of antibiotics. The simplicity and scalability of MOSR indicate its potential for widespread implementation, offering a robust solution to address these critical healthcare issues on a larger scale and ultimately improve patient outcomes.

## Introduction

Emerging infectious diseases and Antimicrobial Resistance (AMR) are two of the most relevant threats to healthcare systems and worldwide society [1]. They contribute to patients' morbidity and mortality [2], and severely increase the prevalence of poor and adverse outcomes [3,4]. Individuals admitted to the Intensive Care Unit (ICU) face a heightened susceptibility to bloodstream infections (BSIs), with a noteworthy incidence rate of 35%. These infections are often attributed to Gram-negative bacteria, particularly those that are resistant to carbapenem antibiotics, and pathogens categorized as DTR (difficult-to-treat and drug-resistant) [5–7]. From the moment a patient is admitted to the ICU with a suspect of sepsis or infection the current protocol focuses on collecting multiple biological samples from the patients for culturing bacteria, a process that typically takes two to three days to yield conclusive results. Meanwhile, clinicians often confront challenging decisions in this context, where they initiate broad-spectrum antibiotics for source control to address the underlying infection. These choices become particularly complex when clinicians must decide on antibiotic initiation while awaiting the results of laboratory tests [8,9] (In Fig 1, we depict a typical timeline).

The widespread or improper utilization of antibiotics can fuel the emergence of AMR [10]. Additionally, improper infection control (IC) measures can further propagate AMR, posing a significant issue in the ICU environment [11] and subjecting patients to elevated risks of increased morbidity [3,4] and mortality [2]. Addressing this issue involves a multifaceted approach that not only encompasses traditional methods like monitoring, pre-emptive measures, swift identification, and the creation of novel medications and immunizations, but also leverages modern advancements. Although cutting-edge rapid solutions for point-of-care diagnostics exist, their widespread adoption is still in progress [5].

As a result, research efforts are also geared towards identifying new correlations in data available upon admission, while waiting for lab results, to develop more universally applicable models. The forecasts of BSI and AMR has recently gained prominence due to the rise of machine learning (ML) techniques and the availability of extensive clinical datasets [12]. Though BSI and AMR are significant concerns, they are statistically rare or rare events. This rarity inherently complicates their detection and management using traditional ML models, which often lack the required specificity and sensitivity for such tasks [13,14]. A compelling alternative for tackling these challenges is the application of Multi-Objective Symbolic Regression (MOSR) to medical databases; this approach has yielded promising results, improving both performance and the potential for clinical interpretability [15].

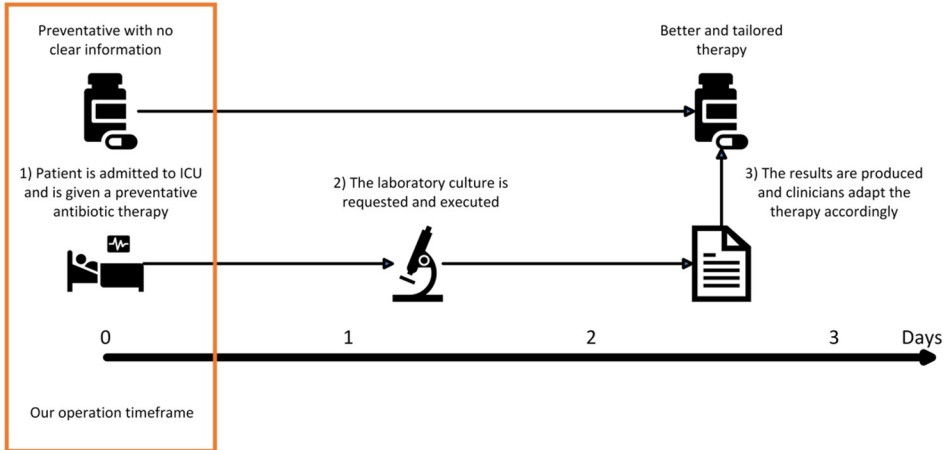

**Fig 1. Timeline of the decision-making process regarding antimicrobial therapy in ICU.**

This retrospective study endeavours to tackle a persistent challenge within the clinical landscape by introducing MOSR, a pioneering ML algorithm that has proven to be very effective in handling naturally imbalanced datasets [15]. Particularly, the choice of MOSR lies on its multi-objective capabilities and the ability to induce the model's training to optimize also the F1-Score together with standard Binary Crossentropy. By choosing MOSR as ML algorithm we expect to replicate the same behaviour seen in literature and obtain a set of models able to produce more balanced performance metrics and put more attention in the least represented of the two classes.

Our focus lies in reshaping the conventional understanding of early detection of AMR and BSI in a clinical context, leveraging routinely accessible EHR data such as blood biomarkers, ongoing pharmacological interventions, and established medical histories. In contrast to prevailing ML methodologies tailored for tabular data, our approach seeks to underscore the inherent limitations in addressing the complexities of this clinical scenario. By delineating two interconnected tasks—first, discerning the presence or absence of BSI, and second, establishing the correlation between infections and AMR—we aim to underscore the inadequacies of traditional ML methods in addressing this intricate issue. This study not only highlights the existence of a persisting problem but also pioneers a transformative perspective, proposing a viable path forward that redefines our approach to such challenges for the first time.

## Methods

### Ethics statement

Clinical, demographic, microbiological, antimicrobial treatment, intervention, bed occupancy, and staffing level data were extracted from intensive care (CareVue; Philips), microbiology (MC&S; GSTT), and electronic patient administration systems (iSoft) to form the anonymized Guy's and St Thomas' Staphylococcal Transmission and Antimicrobial Record database, with approval from the hospital ethics committee (10/H1102/80). Access to the data is subject to appropriate checks and vetting and researchers can refer to the contact of the database manager Finola Higgins (finola.higgins@kcl.ac.uk).

This research was funded/supported by King's College London, DRIVE-Health, KCL funded Centre for Doctoral Training (CDT) in Data-Driven Health, and Guy's and St. Thomas' NHS Fundation Trust. The views expressed are those of the author(s) and not

necessarily those of the NHS, the NIHR or the Department of Health and Social Care. The authors acknowledge the use of the King's College London CREATE computing infrastructure [16] to run the experiments in the Trusted Research Environment.

## Dataset description

Initially, our dataset encompassed over 5000 heterogeneous patients of Guy's and St. Thomas' Hospitals in London admitted to the ICU from Admission and Emergency, hospital wards or operating theatre, with data collected at the patients' admission. The data collection spans 8 years and encompass all type of complex patients who are taken care of in a high complexity university hospital. However, in the interest of data completeness, we excluded individuals with missing values in any of the selected variables. Consequently, our final dataset consists of 1142 patients, and Table 1 provides an overview of variable distributions categorized by one of the primary outcomes, which is BSI.

As we selected non-linear ML models, we opted for a non-linear features selection approach. Namely, we used Hilbert-Schmidt Independence Criterion Lasso (HSIC Lasso) [17] to find the minimal and optimal subset of features to explain a given phenomenon. Only the features having a HSIC score larger than $10^{-3}$ are used in the ML algorithms. The HSIC Lasso

**Table 1. Description of the dataset.**

|  | Feature | Total | BSI—False | BSI—True | p-Value |
|---|---|---|---|---|---|
|  | Population | 1142 (100%) | 1010 (88.44%) | 132 (11.56%) |  |
| APACHE-II Score components | Age Points | 3.35 (2.22) | 3.33 (2.24) | 3.44 (2.11) | = 0.61 |
|  | Chronic Health Points | 0.56 (1.61) | 0.55 (1.6) | 0.6 (1.72) | = 0.76 |
|  | Mean Arterial Pressure Points | 1.99 (0.95) | 1.96 (0.93) | 2.16 (1.06) | = 0.02 |
|  | Oxygenation Points | 1.57 (1.48) | 1.56 (1.47) | 1.62 (1.52) | = 0.64 |
| Laboratory Data | Albumin | 25.44 (8.3) | 25.42 (8.25) | 25.6 (8.74) | = 0.81 |
|  | Bicarbonate | 20.93 (5.07) | 21.2 (4.93) | 18.89 (5.67) | < 0.01 |
|  | Bilirubin | 26.76 (55.9) | 26.02 (57.01) | 32.42 (46.35) | = 0.21 |
|  | Glucose | 7.28 (5.41) | 7.25 (5.26) | 7.52 (6.48) | = 0.59 |
|  | Haematocrit | 30.92 (6.04) | 30.87 (5.95) | 31.32 (6.72) | = 0.42 |
|  | International Normalized Ratio | 1.45 (0.63) | 1.43 (0.62) | 1.54 (0.7) | = 0.07 |
|  | PaO$_2$ | 12.94 (6.18) | 12.83 (5.93) | 13.81 (7.79) | = 0.08 |
|  | Potassium | 4.12 (1.19) | 4.11 (1.19) | 4.21 (1.17) | = 0.32 |
|  | Glutamic-oxaloacetic Transaminase (SGOT) | 121.01 (339.06) | 115.9 (314.56) | 160.07 (487.27) | = 0.15 |
|  | Sodium | 140.72 (8.38) | 140.88 (8.3) | 139.5 (8.9) | = 0.07 |
|  | Urea | 12.57 (9.86) | 12.36 (9.53) | 14.16 (12.03) | = 0.04 |
|  | Urine volume in 24h | 1809.44 (1563.64) | 1825.11 (1554.35) | 1689.52 (1634.14) | = 0.34 |
|  | White Blood Cells | 43.55 (980.57) | 14.67 (8.51) | 264.49 (2884.18) | < 0.01 |
| Medical History | Circulatory System Diseases, No | 1103 [96.58%] | 978 [96.83%] | 125 [94.7%] | = 0.31 |
|  | Circulatory System Diseases, Yes | 39 [3.42%] | 32 [3.17%] | 7 [5.3%] |  |
| Signs and Symptoms | Heart rate per minute | 96.7 (34.32) | 96.77 (34.5) | 96.19 (33.02) | = 0.85 |
|  | Mean Blood Pressure | 68.77 (23.69) | 69.56 (24.06) | 62.67 (19.7) | < 0.01 |
|  | Temperature | 37.84 (1.69) | 37.9 (1.63) | 37.36 (2.01) | < 0.01 |
| Therapies and Treatments | Number of ABs + AVs + AFs per patient * | 2.3 (1.13) | 2.33 (1.12) | 2.05 (1.19) | < 0.01 |
|  | Chlorhexidine, No | 478 [41.86%] | 426 [42.18%] | 52 [39.39%] | = 0.60 |
|  | Chlorhexidine, Yes | 664 [58.14%] | 584 [57.82%] | 80 [60.61%] |  |

* Antibiotics (AB), Antivirals (AV), and Antifungals (AF)

method finds the optimal features subset solving a feature-wise kernelized Lasso optimization problem with a non-negative constraint. We use the same feature set to train all ML algorithms in the comparison.

HSIC Lasso reduces the number of variables from 100 to 25. These include individual components of the APACHE-II score [18], laboratory data, symptoms, past medical history, and medical therapies that patients were undergoing at the time of admission to the ICU. Circulatory system diseases were defined with the ICD-9 code [19] between 390 and 459. These variables are usually available at patient admission as they do not rely on previous health records but reflect the patient's status.

In our study, we model BSI and AMR as binary outcomes due to the limited quantity and representation of available data, which prevented us from delving deeper into individual organism details. BSI is defined as a positive blood culture, whereas AMR is a positive blood culture presenting resistance to at least one antimicrobial medicines. We did not assess the effectiveness of the treatment before the blood culture because the samples for the cultures were collected at the time of admission, which did not allow for a proper evaluation of the efficacy of the AB therapy.

While it is indeed the first instance of predicting BSI and AMR from readily available patient variables, we prioritize safety and knowledge generation. To this end, we selected and compared the following state-of-the-art algorithms: Logistic Regression (LR), Decision Trees (DT), Random Forests (RF), Gradient Boosting (GB), Extreme Gradient Boosting (XGBoost), Light Gradient Boosting (LightGBM), Linear and Quadratic Discriminant analyses (LDA and QDA), Extra Tree (ET), AdaBoost, and Symbolic Regression (SR). We did not use Deep Learning strategies because we aimed at algorithms that would provide easily understandable interpretation, either intrinsic (like in LR or SR) or post hoc (like SHAP [20] or Lime [21] for tree-based methods). For all algorithm, a comprehensive grid search optimization of hyperparameters has been conducted during training.

## Symbolic regression

Symbolic regression is an evolutionary algorithm that generates predictive models in the shape of tree expressions where internal nodes are operations while leaves are features or constants [22]. The algorithm starts with a group of P initial random formulas, i.e., the initial population. At each training step, it generates new individuals by applying random mutations, and then it keeps the best P individuals based on their predictive performance.

The SR method [23,24] was shown to provide exceptional management of naturally imbalanced data [22,25] with the ability to model highly non-linear relations with limited data both in terms of data points and features available. Because the algorithm can choose a subset of variables, it can also be considered a feature selection approach itself, laying further potential for its use in the healthcare domain. Contrary to other non-linear ML algorithms, SR generates flexible mathematical equations that use a customizable set of mathematical operations and feature interactions. In this work, we considered: sum, multiplication, exponential, logarithm, square root, power, and minimum/maximum between two numbers. Although SR is the least common and documented approach, its use has been recently revived in clinical applications for the remarkable potential of a multi-objective training setup [22,25].

MOSR can simultaneously optimize multiple performance measures, also called fitness functions. Its optimisation can produce several equally optimal models, identified through dominance criteria, that create the so-called first Pareto front. We used a MOSR implementation available on GitHub [26] that uses the Non-Dominant Sorting Algorithm II genetic algorithm (NSGA-II) [27].

The optimal formulas show different properties and qualities and allow domain experts to choose the most appropriate model. The flexible and usually compact models arising from SR are naturally interpretable and allow directly inspecting the relations between the variables and identifying the most relevant ones used in the prediction. These features are very useful in a clinical environment as they allow for maintaining control of the model complexity (e.g., limiting non-linearities or setting a maximum length). In this experiment we optimize Binary Crossentropy (BCE), F1 Score and the complexity of the model (how many operations and variables are used).

The clinical implication of an equation generated by MOSR is still an open research challenge and necessitates of further methodological exploration.

From the technical implementation point of view, particularly thinking about Clinical Decision Support Systems (CDSS) software with Electronic Health Records (EHR), one of the unique features of MOSR is that models are plain expressions that can be easily transferred to any information system (even copied and pasted in a spreadsheet) and therefore their technical implementation is much simpler than any other ML alternative. The precondition to achieve this seamless integration would be like any other ML approach, that is producing consistent data preprocessing to feed as input to the model. The challenge of prediction representation to support clinical decision making would a decision in and of itself like in all other ML alternatives as it strongly depends on whether the output should be an action instruction (e.g. "prescribe Enalapril") or more of an additional piece of information for a wider diagnostic reasoning (e.g., "Risk of death is 5%").

## Experimental design and statistics

We partitioned the dataset into training (80%) and test (20%) sets, ensuring stratification based on the class frequency due to the substantial class imbalance between BSI and non-BSI cases. We trained our ML models under two distinct conditions: firstly, without balancing the training set, allowing us to assess the ML algorithm's performance in real-world scenarios, and secondly, by preprocessing the training set solely using the SMOTE [28] method to oversample the minority class, enabling us to evaluate how ML algorithm performance changes under ideal training conditions. The test dataset was unaltered and imbalanced to validate the ML models. For each ML algorithm, we conducted hyperparameter tuning via a standard grid search method and incorporated class weighting, assigning each class a weight equivalent to the inverse of its prevalence in the dataset. Moreover, for each ML model we tuned its hyperparameters using a standard grid search for the best predictive performance.

In the context of our binary classification task for both outcomes, our training prioritizes conservative models, which prioritize high Sensitivity while accepting a penalty in Specificity. Specifically, we place a greater emphasis on minimizing FNs, as these errors correspond to patients with BSI or AMR who may be overlooked, posing significant risks for the patient. To enhance the models discussed in this study, future efforts should focus on training on larger datasets and seeking ways to augment their Specificity, thus ensuring their reliability for deployment in real-world clinical scenarios.

## Results

### Epidemiological analysis

This retrospective study uses data of 1142 patients collected from the ICUs within Guy's and St. Thomas' NHS Foundation Trusts in Central London during this specific period, we conducted a detailed examination of this intriguing phenomenon. All patients of at least 18 year of age at their admission to ICU are included in this dataset.

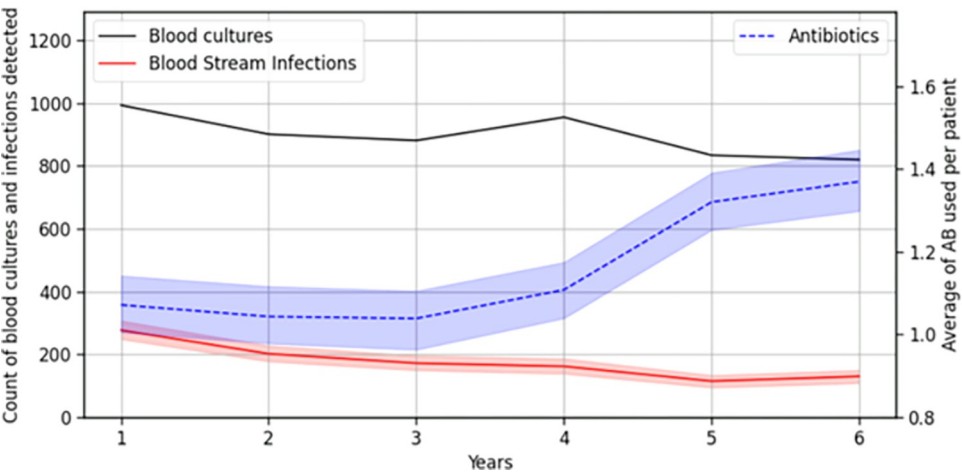

**Fig 2. Evolution of the number of blood cultures, BSI detected, and the average use of AMs used per patient.**

Fig 2 reports data from the study and reveals that although the number of requested blood cultures decreased by a modest 17%, the count of BSI experienced a substantial drop of over 50% during the same timeframe. Furthermore, antibiotics (ABs) usage per patient demonstrated an increase of more than 27%. Notably, the correlation between the number of blood cultures requested and the infections detected showed a high positive correlation of 0.84 ($p<0.05$), aligning with expectations. In contrast as the rate of ABs usage increased, the infections detected decreased, and vice versa. This was shown by a negative correlation of -0.70 ($p<0.05$). These results indicated that the infections detected, and the rate of ABs usage moved in opposite directions, highlighting a discrepancy between the actual therapeutic need for antibiotics and their utilization for the time analysed.

## Data description

In Table 1, we provide a description of the selected variables and conduct a comparison of their distribution among groups categorized by the presence or absence of BSI. The statistical relevance is assessed by means of the appropriate statistical test based on the data type and distribution (t-test for the continuous variables, ANOVA for categorical variables). It is worth noting that most variables exhibit non-statistically significant differences between the two groups, primarily due to the limited biological relevance of these variables to the outcome. Nevertheless, these variables are retained in our experiments because certain non-linear methods, such as SR, have the capacity to generate complex features that are functions of the original variables. These newly created composite features may reveal statistically distinct distributions among groups of patients. The presence or absence of such non-linearities cannot be ruled out solely by comparing feature distributions.

The class imbalance that manifest in this analysis is representative of the general population and intrinsic behaviour of many clinical phenomena. This paper intends to not alter this distribution and propose a ML methodology able to overcome this behaviour better than state-of-the-art approaches, therefore enabling a new way of tackling this class of clinical prediction tasks.

## MOSR performance compared to state-of-the-art ML algorithms

Tables 2 and 3 showcase the performance metrics assessed on the test dataset, encompassing BSI and AMR detection respectively, across various ML approaches. We examine both

**Table 2. Detection of BSI: predictive performance on test dataset.**

| | Real-world imbalanced training dataset | | | | | | | SMOTE re-balanced training dataset | | | | | | |
|---|---|---|---|---|---|---|---|---|---|---|---|---|---|---|
| | AUC | Acc. | Prec. | Spec. | Sens. | F1 | Feat. | AUC | Acc. | Prec. | Spec. | Sens. | F1 | Feat. |
| LR | .61 | .88 | .17 | .00 | .03 | .05 | 24 | .57 | .60 | .14 | .00 | .48 | .22 | 24 |
| DT | .48 | .78 | .08 | .89 | .09 | .08 | 24 | .52 | .75 | .15 | .82 | .22 | .17 | 24 |
| RF | .60 | .88 | .10 | .00 | .01 | .02 | 24 | .60 | .88 | .08 | .00 | .02 | .03 | 24 |
| GB | .60 | .88 | .27 | .00 | .05 | .08 | 24 | .56 | .86 | .16 | .00 | .05 | .08 | 24 |
| XGBoost | .59 | .87 | .00 | .00 | .00 | .00 | 24 | .57 | .86 | .22 | .00 | .09 | .12 | 24 |
| LightGBM | .59 | .88 | .01 | .00 | .03 | .01 | 24 | .56 | .86 | .17 | .00 | .03 | .05 | 24 |
| LDA | .58 | .88 | .06 | .00 | .02 | .03 | 24 | .57 | .62 | .15 | .00 | .52 | .24 | 24 |
| QDA | .55 | .88 | .26 | .00 | .04 | .06 | 24 | .48 | .84 | .18 | .00 | .06 | .07 | 24 |
| ET | .59 | .88 | .00 | .00 | .00 | .00 | 24 | .55 | .88 | .35 | .00 | .04 | .07 | 24 |
| AdaBoost | .55 | .86 | .20 | .00 | .11 | .14 | 24 | .53 | .80 | .16 | .00 | .16 | .16 | 24 |
| MOSR Best AUROC | .79 | .75 | .79 | .69 | .27 | .39 | 3 | .72 | .26 | .17 | .92 | .12 | .22 | 2 |
| MOSR Best F1 | .69 | .83 | .87 | .54 | .35 | .42 | 6 | .67 | .79 | .83 | .46 | .26 | .33 | 22 |

scenarios of training datasets, one with balanced class distribution and the other with imbalanced class distribution. In the case of standard ML models, we present the outcomes of the performance calculated on the test dataset. For the MOSR case, we provide two key models: one for the model with the highest AUROC score and one for the model yielding the highest F1 score. Highlighting the model with best F1 score in the table is crucial because it offers a valuable measure of model performance, especially in scenarios where precision and recall need to be balanced, making it essential for tasks like BSI and AMR detection where both false positives and false negatives carry different, yet significant consequences.

In S1 Appendix, more details on the models and the confusion matrices are reported.

Our results indicate that traditional ML approaches struggle to efficiently predict BSI or AMR presence in ICU patients using standard EHR data, while MOSR emerges as a promising alternative. MOSR consistently achieves the highest AUROC and F1, remaining stable in both

**Table 3. Detection of AMR: predictive performance on test dataset.**

| | Real-world imbalanced training dataset | | | | | | | SMOTE re-balanced training dataset | | | | | | |
|---|---|---|---|---|---|---|---|---|---|---|---|---|---|---|
| | AUC | Acc. | Prec. | Spec. | Sens. | F1 | Feat. | AUC | Acc. | Prec. | Spec. | Sens. | F1 | Feat. |
| LR | .64 | .90 | .10 | .00 | .01 | .02 | 23 | .56 | .63 | .13 | .00 | .46 | .20 | 23 |
| DT | .57 | .84 | .21 | .88 | .23 | .22 | 23 | .53 | .78 | .12 | .84 | .21 | .15 | 23 |
| RF | .66 | .90 | .00 | .00 | .00 | .00 | 23 | .61 | .90 | .35 | .00 | .05 | .09 | 23 |
| GB | .61 | .90 | .18 | .00 | .05 | .08 | 23 | .60 | .89 | .29 | .00 | .09 | .10 | 23 |
| XGBoost | .63 | .90 | .05 | .00 | .01 | .02 | 23 | .56 | .90 | .25 | .00 | .05 | .08 | 23 |
| LightGBM | .58 | .90 | .10 | .00 | .01 | .03 | 23 | .57 | .90 | .25 | .00 | .05 | .08 | 23 |
| LDA | .64 | .90 | .28 | .00 | .05 | .08 | 23 | .57 | .66 | .13 | .00 | .46 | .20 | 23 |
| QDA | .46 | .89 | .22 | .00 | .10 | .14 | 23 | .41 | .44 | .08 | .00 | .49 | .14 | 23 |
| ET | .65 | .90 | .00 | .00 | .00 | .00 | 23 | .60 | .90 | .30 | .00 | .04 | .07 | 23 |
| AdaBoost | .56 | .89 | .23 | .00 | .11 | .14 | 23 | .55 | .85 | .17 | .00 | .13 | .14 | 23 |
| MOSR Best AUROC | .86 | .78 | .29 | .29 | .91 | .44 | 4 | .85 | .35 | .78 | .13 | .99 | .23 | 11 |
| MOSR Best F1 | .86 | .78 | .29 | .29 | .91 | .44 | 4 | .64 | .82 | .86 | .26 | .45 | .33 | 16 |

imbalanced and balanced training scenarios. Even though SMOTE enhances some metrics, it falls short of ensuring balancing in predictive performance. The improved performance of MOSR on the test sets are an indication of its ability to mitigate overfitting behavior. In summary, MOSR outperforms other ML algorithms in both balanced and imbalanced training datasets, holding potential for predictive modelling in this critical healthcare context.

The MOSR chosen individual shows much higher, balanced, and thus reliable performances with .42 and .44 F1-Scores respectively compared to the next best alternative of DT ad ADA respectively. This indicates that the multi-objective training of MOSR on BCE, AUROC, and F1 was successful.

The MOSR approach is characterized by innovation, yet it could be further enhanced through the incorporation of established machine learning methodological practices.

External validation of MOSR models would be required for a complete assessment of this approach and more experimentation will be needed to further assess the full potential of the multi-objective training. We leave the problem of choosing the best MOSR model for clinical purposes for future work. In fact, only once more extensive and representative datasets are explored will it be appropriate to dive into the details of model choice and explanation. Nonetheless, our work remains a solid proof-of-concept putting light on the way ahead for AMR and AB stewardship research.

## Discussion and conclusion

In summary, the challenges posed by BSI and AMR detection persist as formidable issues within healthcare systems, giving rise to the inappropriate use of antibiotics and unfavourable patient outcomes. This matter is particularly critical in high-risk settings like the ICU, where vulnerable patients are exposed to heightened risks of morbidity and mortality. ICU clinicians make daily decisions regarding antibiotic therapy, often in the absence of conclusive bacterial culture results at the time of admission.

Throughout the period we observed, a significant development took place, wherein the implementation of the IC initiative led to a decrease in infection rates. Paradoxically, this reduction in infections coincided with an uptick in the use of antibiotics per patient. This unexpected divergence in the relationship between antibiotic utilization and infection prevalence at the hospital level prove the persistent need of a rapid diagnostic support. Despite the considerable passage of time and the routine incorporation of the IC program, the approach to managing ICU admissions, encompassing infection detection technology and clinical protocols, has remained largely unaltered paving the way for the implementation of a clinical decision support tool like ours.

In this study, advanced ML models was used to predict the occurrence of BSI and AMR based on variables already available for every patient at ICU admission. Our models outperformed classical ML models and demonstrated promise for further ML development in this field. Our dataset reflects an era where the technological and behavioural approaches to AM stewardship in ICU admissions have remained largely unchanged. In this context, our findings reveal that MOSR emerges as the most promising approach for BSI and AMR detection, utilizing commonly available Electronic Health Record (EHR) data and without relying on previous microbiological data relating infections and resistances. These results hold substantial implications for improving early diagnosis and targeted AM therapy, ultimately contributing to the ongoing battle against the growing threat of AMR in healthcare settings.

A comprehensive literature search on Pubmed, Google Scolar and Scopus has produced a selection of seven papers that approach the problem of AMR prediction using EHR and ML

techniques. Traditional approaches to identifying AMRs use AM susceptibility testing based on phenotypic testing [29] while more recent methods use ML and deep learning to analyse genome sequences directly [30,31], including Lewin-Epstein et al. [32], where an ensemble of ML models is used to predict the resistance to five ABs. However, all these approaches start from bacterial cultures data and not EHR data that are commonly available at ICU admission in a predictive manner. Looking at EHR-focused efforts, a study by Moran et al. [33] used EHRs to predict resistance to co-amoxiclav and piperacillin/tazobactam with AUC ranging from .61 to .67. Their predictive analysis relies on previously acquired blood cultures. Garcia-Vidal et al. [34] show how their ML approach can use EHRs to predict multidrug-resistant Gram-negative bacilli with AUC ranging from .79 to .97. The chosen variables may be hard to obtain as they rely on an extensive past medical history that is not often available and previous blood cultures. Moreover, although the number of records they used is 3235, the number of patients is only 349, making the model very much prone to overfitting. Pascual-Sanchez et al. [35], study a similar application and strategy, using a single data point per patient re-balancing the outcomes by under-sampling the records not presenting AMR in the final dataset. Overall, the reviewed approaches make use of variables that are not readily available at patient admission in their EHR records, which was a core study design choice of our proposal. Moreover, this work is the first of its kind proposing the introduction of the methodology of MOSR and multi-objective training.

To the best of our knowledge, this is the first work aiming to predict the outcome of bacterial cultures using easily and readily collectible EHR data in a competitive way. This makes our result an exciting and appealing front-line tool for clinicians waiting for laboratory evidence. In our future work, we will focus on improving our models' robustness, reliability, and interpretability to facilitate their use in real-world situations. Moreover, collecting new data would allow extending the current approach to individual microorganism AMR detection and sharpening the algorithm performance.

This study, while valuable, is constrained by several limitations. The first is data scarcity, with our dataset comprising approximately 1000 patients. This limited size may affect the robustness and generalizability of our findings. A possible solution to this limitation is the use of Federated Learning strategies to allow more clinical institutions to participate in the training process with their data.

The second limitation is data imbalance, particularly in binary outcomes such as BSI and AMR. This disparity can introduce bias and affect the performance of ML models. Despite this, our approach offers a promising solution for imbalanced prediction tasks.

The third limitation relates to the need to identify the most effective and reliable representation of the predictions for a proper integration with clinical decision support systems. This would require a closer work with practicing clinician to understand the nuances of their decision-making environment and what type of representation would convey the prediction best.

Lastly, the absence of external validation limits the broader applicability of our approach. However, the primary focus of this work is to propose a novel method for addressing imbalanced prediction tasks in healthcare, rather than resolving clinical challenges.

To address these limitations, we are actively working on expanding our dataset, both in terms of size and diversity, to enhance the representativeness of our findings. Additionally, we are reevaluating our approach to potentially explore predictions at a finer level, focusing on individual organisms rather than generic binary outcomes. These efforts aim to overcome the data limitations, refine our methodology, and ultimately provide more comprehensive insights into the complex dynamics of healthcare-associated infections and AMR.

## Supporting information

**S1 Appendix. Models and confusion matrices.** When the equations are of reasonable complexity, we report it below, otherwise we omit them in the manuscript. Variables need to be scaled between 0 and 1.
(DOCX)

## Author Contributions

**Conceptualization:** Davide Ferrari, Pietro Arina, Jonathan Edgeworth, Vasa Curcin, Veronica Guidetti, Yanzhong Wang.

**Data curation:** Davide Ferrari.

**Formal analysis:** Davide Ferrari.

**Funding acquisition:** Jonathan Edgeworth, Vasa Curcin, Yanzhong Wang.

**Investigation:** Davide Ferrari.

**Methodology:** Davide Ferrari, Pietro Arina, Veronica Guidetti, Federica Mandreoli.

**Project administration:** Davide Ferrari.

**Resources:** Davide Ferrari.

**Software:** Davide Ferrari, Veronica Guidetti.

**Supervision:** Jonathan Edgeworth, Vasa Curcin, Federica Mandreoli, Yanzhong Wang.

**Validation:** Davide Ferrari.

**Visualization:** Davide Ferrari.

**Writing – original draft:** Davide Ferrari, Pietro Arina.

**Writing – review & editing:** Davide Ferrari, Pietro Arina, Jonathan Edgeworth, Vasa Curcin, Veronica Guidetti, Federica Mandreoli, Yanzhong Wang.

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
