## [Decision Letter · Decision Letter 0]

3 Jan 2024

PDIG-D-23-00418

Using interpretable Machine Learning to predict bloodstream infection and antimicrobial resistance in patients admitted to ICU: early alert predictors based on EHR data to guide antimicrobial stewardship

PLOS Digital Health

Dear Dr. Ferrari,

Thank you for submitting your manuscript to PLOS Digital Health. After careful consideration, we feel that it has merit but does not fully meet PLOS Digital Health's publication criteria as it currently stands. Therefore, we invite you to submit a revised version of the manuscript that addresses the points raised during the review process.

Please submit your revised manuscript within 60 days Mar 03 2024 11:59PM. If you will need more time than this to complete your revisions, please reply to this message or contact the journal office at digitalhealth@plos.org. Please include the following items when submitting your revised manuscript:

We look forward to receiving your revised manuscript.

Kind regards,

Akhilanand Chaurasia

Section Editor

PLOS Digital Health

Journal Requirements:

1. We ask that a manuscript source file is provided at Revision. Please upload your manuscript file as a .doc, .docx, .rtf or .tex.

2. Please provide separate figure files in .tif or .eps format only and remove any figures embedded in your manuscript file. Please also ensure that all files are under our size limit of 10MB.

3. Please insert an Ethics Statement at the beginning of your Methods section, under a subheading 'Ethics Statement'. It must include:

1) The name(s) of the Institutional Review Board(s) or Ethics Committee(s)

2) The approval number(s), or a statement that approval was granted by the named board(s) 

3) (for human participants/donors) - A statement that formal consent was obtained (must state whether verbal/written) OR the reason consent was not obtained (e.g. anonymity). NOTE: If child participants, the statement must declare that formal consent was obtained from the parent/guardian.

Additional Editor Comments (if provided):

Dear Author,

Kindly provide satisfactory rebuttal of queries raised by reviewers

Reviewers' comments:

Reviewer's Responses to Questions

**Comments to the Author**

1. Does this manuscript meet PLOS Digital Health’s publication criteria? Is the manuscript technically sound, and do the data support the conclusions? The manuscript must describe methodologically and ethically rigorous research with conclusions that are appropriately drawn based on the data presented.

Reviewer #1: Partly

Reviewer #2: Yes

Reviewer #3: Yes

2. Has the statistical analysis been performed appropriately and rigorously?

Reviewer #1: No

Reviewer #2: Yes

Reviewer #3: N/A

3. Have the authors made all data underlying the findings in their manuscript fully available (please refer to the Data Availability Statement at the start of the manuscript PDF file)?

Reviewer #1: Yes

Reviewer #2: Yes

Reviewer #3: No

4. Is the manuscript presented in an intelligible fashion and written in standard English?

Reviewer #1: Yes

Reviewer #2: No

Reviewer #3: Yes

5. Review Comments to the Author

Reviewer #1: The article presents a study that introduces a machine learning approach based on Multi-Objective Symbolic Regression (MOSR) for predicting bloodstream infections (BSI) and antimicrobial resistance (AMR) in ICU patients using electronic health record (EHR) data. The study compares MOSR with traditional machine learning models, highlighting its superior performance in minimizing false negatives and its potential for large-scale implementation to guide antimicrobial stewardship. There are several issues with this article that need improvement:

• The binary outcomes for BSI and AMR can introduce bias in the machine learning models due to class imbalance.

• The study lacks external validation of the MOSR models, which is necessary for assessing their real-world applicability.

• While the study emphasizes the interpretability of MOSR, the complexity of the models used, and their interpretability in a clinical setting are not thoroughly addressed.

• The study's reliance on a relatively small and specific dataset might not adequately represent the diverse scenarios encountered in different healthcare settings.

• Concerns about the class imbalance and lack of external validation might limit confidence in the study’s findings.

• The study does not sufficiently address how its findings can be practically implemented in real-world clinical settings, especially considering the complexity of the models.

• The work should expand the dataset to enhance representativeness, address data limitations, refine methodology, and improve model robustness and interpretability for practical clinical use.

Reviewer #2: I don't have access to specific questions or answers related to a review. If you have any specific questions or content related to a review that you'd like assistance with, please provide more details, and I'll do my best to help.

Reviewer #3: The submitted manuscript aims to analyze and predict bloodstream infections (BSI) and antimicrobial resistance (AMR) using machine learning (ML) techniques including Logistic Regression, Decision Trees, Random Forests, and symbolic regression.

I have few points to raise:

1. The definition of AMR, authors stated that the definition of AMR line 111, “AMR was the culture presenting at least one resistance to ABs.” I don’t agree with definition, it should be more like “AMR was the culture presenting at least one resistance to Abs that once was successful treating the microbe”.

2. Please provide more data regarding AMR if available, the frequency of AMR is quiet high , 11 compared to 13 BSI in the test data reflecting high resistance that could be the definition used. 

3. Please provide the 25 features used in the analysis in the appendix.

4. Is there any reason the authors choose this split training 90%, test 10% which deviates from normal ML approaches.

5. Looking at the symbolic regression equations, it seems the variables are interesting and the authors may provide explanation on them, example biliribuin reflecting liver injury, urea reflecting kidney injury along with temperature and mean arterial pressure, but my main concern is the number of antibiotics and the use of nystatin which primarily reflect higher clinical suspicious of BSI and AMR rather then actual predictive variable as it reflects the physician response to the presentation rather then the presentation itself. Alternative more suitable variable would be recent antibiotic use before the presentation.

6. PLOS authors have the option to publish the peer review history of their article (what does this mean?). If published, this will include your full peer review and any attached files.

**Do you want your identity to be public for this peer review?** For information about this choice, including consent withdrawal, please see our Privacy Policy.

Reviewer #1: Yes: Shadab Alam

Reviewer #2: No

Reviewer #3: No

---

## [Decision Letter · Decision Letter 1]

24 May 2024

PDIG-D-23-00418R1

Using interpretable Machine Learning to predict bloodstream infection and antimicrobial resistance in patients admitted to ICU: early alert predictors based on EHR data to guide antimicrobial stewardship

PLOS Digital Health

Dear Dr. Ferrari,

Thank you for submitting your manuscript to PLOS Digital Health. After careful consideration, we feel that it has merit but does not fully meet PLOS Digital Health's publication criteria as it currently stands. Therefore, we invite you to submit a revised version of the manuscript that addresses the points raised during the review process.

Please submit your revised manuscript within 30 days Jun 23 2024 11:59PM. If you will need more time than this to complete your revisions, please reply to this message or contact the journal office at digitalhealth@plos.org. Please include the following items when submitting your revised manuscript:

We look forward to receiving your revised manuscript.

Kind regards,

Akhilanand Chaurasia

Section Editor

PLOS Digital Health

Journal Requirements:

2. Please insert an Ethics Statement at the beginning of your Methods section, under a subheading 'Ethics Statement'. It must include:

1) The name(s) of the Institutional Review Board(s) or Ethics Committee(s)

2) The approval number(s), or a statement that approval was granted by the named board(s) 

[3) (for human participants/donors) - A statement that formal consent was obtained (must state whether verbal/written) OR the reason consent was not obtained (e.g. anonymity). NOTE: If child participants, the statement must declare that formal consent was obtained from the parent/guardian.]

Additional Editor Comments (if provided):

Dear Author,

Kindly provide satisfactory rebuttal of queries raised by reviewers

Reviewers' comments:

Reviewer's Responses to Questions

**Comments to the Author**

1. If the authors have adequately addressed your comments raised in a previous round of review and you feel that this manuscript is now acceptable for publication, you may indicate that here to bypass the “Comments to the Author” section, enter your conflict of interest statement in the “Confidential to Editor” section, and submit your "Accept" recommendation.

Reviewer #2: All comments have been addressed

Reviewer #3: (No Response)

2. Does this manuscript meet PLOS Digital Health’s publication criteria? Is the manuscript technically sound, and do the data support the conclusions? The manuscript must describe methodologically and ethically rigorous research with conclusions that are appropriately drawn based on the data presented.

Reviewer #2: Partly

Reviewer #3: (No Response)

3. Has the statistical analysis been performed appropriately and rigorously?

Reviewer #2: N/A

Reviewer #3: Yes

4. Have the authors made all data underlying the findings in their manuscript fully available (please refer to the Data Availability Statement at the start of the manuscript PDF file)?

Reviewer #2: Yes

Reviewer #3: Yes

5. Is the manuscript presented in an intelligible fashion and written in standard English?

Reviewer #2: Yes

Reviewer #3: Yes

6. Review Comments to the Author

Reviewer #2: Could you clarify the advantages of the MOSR approach over conventional machine learning techniques in diagnosing bloodstream infections and Antimicrobial Resistance? Furthermore, what might be the ramifications of this method in clinical practice and patient outcomes?

The MOSR approach is characterized by innovation, yet it could be further enhanced through the incorporation of established machine learning methodological practices, such as validation, class imbalance mitigation, and clinical evaluation

Reviewer #3: Authors' responses should be comprehensive and integrated into the manuscript (with a reference to the location of adjustment), rather than merely being addressed in separate replies.

7. PLOS authors have the option to publish the peer review history of their article (what does this mean?). If published, this will include your full peer review and any attached files.

**Do you want your identity to be public for this peer review?** For information about this choice, including consent withdrawal, please see our Privacy Policy. 

Reviewer #2: No

Reviewer #3: No

---

## [Decision Letter · Decision Letter 2]

12 Sep 2024

Using interpretable Machine Learning to predict bloodstream infection and antimicrobial resistance in patients admitted to ICU: early alert predictors based on EHR data to guide antimicrobial stewardship

PDIG-D-23-00418R2

Dear Mr Ferrari,

We are pleased to inform you that your manuscript 'Using interpretable Machine Learning to predict bloodstream infection and antimicrobial resistance in patients admitted to ICU: early alert predictors based on EHR data to guide antimicrobial stewardship' has been provisionally accepted for publication in PLOS Digital Health.

Best regards,

Akhilanand Chaurasia

Section Editor

PLOS Digital Health

Reviewer Comments (if any, and for reference):

Reviewer's Responses to Questions

**Comments to the Author**

1. If the authors have adequately addressed your comments raised in a previous round of review and you feel that this manuscript is now acceptable for publication, you may indicate that here to bypass the “Comments to the Author” section, enter your conflict of interest statement in the “Confidential to Editor” section, and submit your "Accept" recommendation.

Reviewer #3: All comments have been addressed

2. Does this manuscript meet PLOS Digital Health’s publication criteria? Is the manuscript technically sound, and do the data support the conclusions? The manuscript must describe methodologically and ethically rigorous research with conclusions that are appropriately drawn based on the data presented.

Reviewer #3: (No Response)

3. Has the statistical analysis been performed appropriately and rigorously?

Reviewer #3: (No Response)

4. Have the authors made all data underlying the findings in their manuscript fully available (please refer to the Data Availability Statement at the start of the manuscript PDF file)?

Reviewer #3: (No Response)

5. Is the manuscript presented in an intelligible fashion and written in standard English?

Reviewer #3: (No Response)

6. Review Comments to the Author

Reviewer #3: (No Response)

7. PLOS authors have the option to publish the peer review history of their article (what does this mean?). If published, this will include your full peer review and any attached files.

**Do you want your identity to be public for this peer review?** For information about this choice, including consent withdrawal, please see our Privacy Policy.

Reviewer #3: No
